

# Analysis of the latitudinal variability of tropospheric ozone in the Arctic using the large number of aircraft and ozonesonde observations in early summer 2008

Gerard Ancellet[1], Nikos Daskalakis[1], Jean Christophe Raut[1], Boris Quennehen[1], François Ravetta[1], Jonathan Hair[2], David Tarasick[3], Hans Schlager[4], Andrew J. Weinheimer[5], Anne M. Thompson[6], Bryan Johnson[7], Jennie L. Thomas[1], and Katharine S. Law[1]

[1]LATMOS/IPSL, UPMC Univ. Paris 06 Sorbonne Universités, UVSQ, CNRS, Paris, France
[2]NASA Langley Reasearch Center, Hampton, VA, USA
[3]Environment and Climate Change Canada, Downsview, ON, Canada
[4]Institut für Physik der Atmosphäre, DLR, Oberpfaffenhofen, Germany
[5]NCAR, Boulder, CO, USA
[6]NASA/GSFC, Greenbelt, MD, USA
[7]NOAA/Earth System Research Laboratory (ESRL), Boulder, CO, USA

*Correspondence to:* Gerard Ancellet : gerard.ancellet@upmc.fr

**Abstract.** The goal of the paper are to: (1) present tropospheric ozone ($O_3$) climatologies in summer 2008 based on a large amount of measurements, during the International Polar Year when the Polar Study using Aircraft, Remote Sensing, Surface Measurements, and Models of Climate Chemistry, Aerosols, and Transport (POLARCAT) campaigns were conducted (2) investigate the processes that determine $O_3$ concentrations in two different regions (Canada and Greenland) that were thoroughly

studied using measurements from 3 aircraft and 7 ozonesonde stations. This paper provides an integrated analysis of these observations and the discussion of the latitudinal and vertical variability of tropospheric ozone north of 55°N during this period is performed using a regional model (WFR-Chem). Ozone, CO and potential vorticity (PV) distributions are extracted from the simulation at the measurement locations. The model is able to reproduce the $O_3$ latitudinal and vertical variability but a negative $O_3$ bias of 6-15 ppbv is found in the free troposphere over 4 km, especially over Canada.

Ozone average concentrations are of the order of 65 ppbv at altitudes above 4 km both over Canada and Greenland, while they are less than 50 ppbv in the lower troposphere. The relative influence of stratosphere-troposphere exchange (STE) and of ozone production related to the local biomass burning (BB) emissions is discussed using differences between average values of $O_3$, CO and PV for Southern and Northern Canada or Greenland and two vertical ranges in the troposphere: 0-4 km and 4-8 km. For Canada, the model CO distribution and the weak correlation ($< 30\%$) of $O_3$ and PV suggests that stratosphere-

troposphere exchange (STE) is not the major contribution to average tropospheric ozone at latitudes less than 70°N, due to the fact that local biomass burning (BB) emissions were significant during the 2008 summer period. Conversely over Greenland, significant STE is found according to the better $O_3$ versus PV correlation ($> 40\%$) and the higher 75th PV percentile.

A weak negative latitudinal summer ozone gradient -6 to -8 ppbv is found over Canada in the mid troposphere between 4 and 8 km. This is attributed to an efficient $O_3$ photochemical production due to the BB emissions at latitudes less than 65°N, while

STE contribution is more homogeneous in the latitude range 55°N to 70°N. A positive ozone latitudinal gradient of 12 ppbv is



observed in the same altitude range over Greenland not because of an increasing latitudinal influence of STE, but because of different long range transport from multiple mid-latitude sources (North America, Europe and even Asia for latitudes higher than 77°N).

# 1 Introduction

Ozone concentrations are still increasing in many locations in the Northern Hemisphere mostly due to an increase in Asian precursor emissions (Parrish et al., 2012). Since tropospheric ozone is an effective greenhouse gas with a relatively long lifetime its main impact on climate and air quality is within mid-latitudes regions. Several studies have shown that ozone makes also an important contribution to Arctic surface temperature increases due to direct local warming in Arctic as well as heat transport following warming due to ozone at mid-latitudes (Shindell, 2007; Shindell et al., 2009; AMAP, 2015). The Arctic ozone budget still requires better quantification and is complicated by the interplay between the downward transport of stratospheric ozone (Hess and Zbinden, 2013), the removal of boundary layer $O_3$ due to halogen chemistry especially in springtime (Simpson et al., 2007; Abbatt et al., 2012), and the photochemical production due to local sources like boreal forest fires (Stohl et al., 2007; Thomas et al., 2013), $NO_x$ enhancement from snowpack emissions (Honrath et al., 1999; Legrand et al., 2009), local summertime production from peroxyacetyl nitrate (PAN) decomposition (Walker et al., 2012) or ship emissions (Granier et al., 2006). Ozone distributions over North America have been discussed for the spring period at high latitude using the Tropospheric Ozone Production about the Spring Equinox (TOPSE) and Arctic Research of the Composition of the Troposphere from Aircraft and Satellites (ARCTAS) data set in several publications (Browell et al., 2003; Wang et al., 2003; Olson et al., 2012; Koo et al., 2012) showing frequent occurrence of ozone depletion event (ODE) but not extending to the free troposphere, a net photochemical production rate of $O_3$ equal to zero throughout most of the troposphere, while transport from mid-latitude and Stratosphere-Troposphere Exchange (STE) explain the ozone increase with latitude.

For the summer period ozone photochemical production is expected according to the numerous studies conducted at mid-latitudes (Crutzen et al., 1999; Parrish et al., 2012), but little attention was given to the high latitude distribution during this season. During the ARCTAS-B (Jacob et al., 2010) and Polar Study using Aircraft, Remote Sensing, Surface Measurements, and Models of Climate Chemistry, Aerosols, and Transport (POLARCAT) campaigns (Law et al., 2014), many ozone measurements have been carried out over Canada and Greenland from 15 June to 15 July 2008 in addition to regular ozonesonde observation by the Canadian network. This allows a detailed analysis of the ozone regional distribution at high latitudes between 55°N and 90°N and a discussion about the relevant ozone sources driving the summer ozone values. So far the relative influence of the main ozone summer sources was mainly derived from modeling studies, e.g. summer simulations of global model simulation of the ozone source attribution (Wespes et al., 2012; Walker et al., 2012; Monks et al., 2015), or regional modeling of biomass burning case studies for North American fires (Thomas et al., 2013) or Asian fires (Dupont et al., 2012).

In order to interpret the measurements, we use a hemispheric simulation performed using the regional chemical transport model WRF-Chem (Grell et al., 2005; Fast et al., 2006). WRF-Chem predicts simultaneously a meteorological forecast, including the dynamics of the UTLS region, as well as emissions and chemistry to predict ozone (and other trace gas and aerosol)



concentrations. Here, we use the model to investigate dynamics that determine ozone concentrations as a function of latitude, referred to as latitudinal gradients, including stratosphere-troposphere exchanges processes, which can bring high ozone air from the stratosphere into the upper troposphere in the Arctic. In addition, we use the model to compare directly predicted and measured ozone in summer 2008 and use a pollution tracer (i.e. carbon monoxide, CO) to separate air influenced by pollution and subsequent ozone formation via photochemistry from air influenced by stratosphere-troposphere mixing processes.

The objectives of this paper are thus twofold : (i) to establish the summer tropospheric ozone latitudinal variability over Greenland and Canada based on the large number of ozone measurements available in the free troposphere during June/July 20008, and (ii) to explore the role of photochemistry and stratosphere/troposphere exchange in the observed high latitude ozone distribution by extracting the potential vorticity (PV) and CO distribution from a 2-month WRF-Chem regional model simulation. The WRF-Chem simulation also provides the modeled ozone distribution to verify the coherence between the modeled PV and CO with observed ozone distributions. The ozone data set and the WRF-Chem simulation are described in section 2 and 3, respectively. The model and measurement ozone comparison is discussed in section 4, while the latitudinal distribution of ozone, CO and PV are presented in section 5.

## 2 Summer 2008 ozone data set

We use three types of ozone measurements to build the data set considered in this work: airborne lidar, in-situ aircraft ozone analyzer, and electrochemical concentration cell (ECC) ozonesondes. Two airborne ozone DIfferential Absorption Lidars (DIAL) are consideredi: the NASA-DC8 instrument and the ALTO lidar on the French ATR-42. In-situ ozone monitors have been installed onboard 3 different aircraft: the NASA DC-8, the DLR Falcon-20 and the French ATR-42. We also considered 7 ground-based stations where ozonesondes are regularly launched over Canada (latitudes $> 50°N$ in the longitude range between -70°W and -160°W) and over Greenland (latitudes $> 55°N$ in the longitude range between -60°W and -20°W). The times and positions of the June/July 2008 measurements included in our study are given in Table 1 for the aircraft observations and in Table 2 for the ozonesondes. The aircraft flight paths over Canada and Greenland are shown in Fig.1.

The German DLR Falcon-20 was based in Greenland and used an UV absorption instrument (Thermo Environment Instruments, TEI49C) to measure $O_3$ with an uncertainty of $\pm 2$ ppbv ($\pm 5\%$ of the signal) (Schlager et al., 1997; Roiger et al., 2011). The ATR-42 aircraft was also based in Greenland and the ozone measurements are made using a similar instrument (TEI49-103) calibrated against a NIST (National Institute of Standards and Technology) referenced $O_3$ calibrator Model49PS at zero, 250, 500 and 750 ppbv (Marenco et al., 1998). A 4-ppbv negative bias related to a slight $O_3$ loss in the ATR-42 air inlet (e.g. 10% for 40 ppbv and 5% for 80 ppbv) has been corrected in this study. On 14 July 2008, the comparison with the DLR-Falcon at two levels over Greenland near 67°N shows an uncertainty better than 2 ppbv (Fig.2). The American NASA DC-8 (Weinheimer et al., 1994) $O_3$ measurements are made with a chemiluminescence technique, with the instrument calibrated by additions of $O_3$ determined by UV optical absorption at 254 nm. Uncertainties in the DC-8 $O_3$ data are typically $\pm 2$ ppbv ($\pm 5\%$ of the signal). The DC8 flew mainly over Canadian forest fire regions in summer (Jacob et al., 2010). The comparison shows a very good precision but a slight 4 ppbv positive difference with the DLR-Falcon data (Fig.2) observed





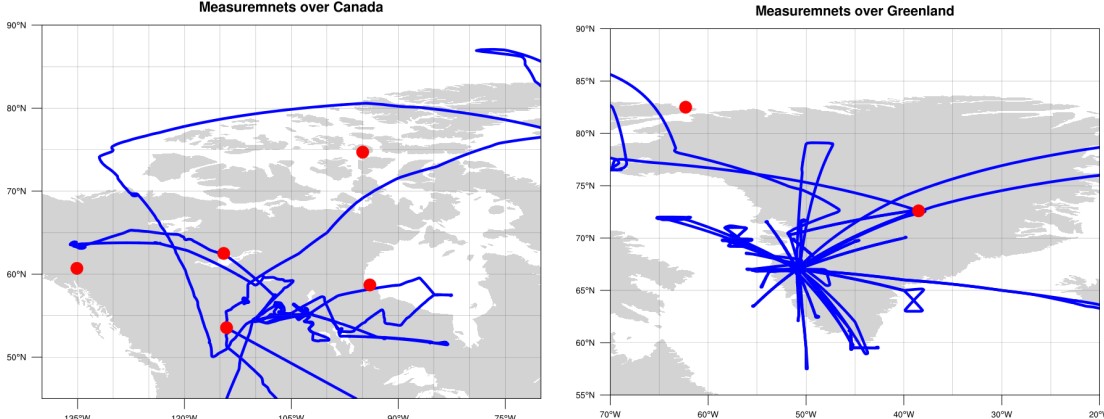

**Figure 1.** Horizontal distribution of the aircraft positions (blue) and sounding locations (red) over the selected Canada (left panel) and Greenland regions (right panel).

during the aircraft intercomparison flight on July 9th over Northern Greenland. None of the DLR-Falcon and NASA DC8 data were corrected.

The ATR-42 ozone DIfferential Absorption Lidar (DIAL) instrument was mounted in a zenith-viewing mode making ozone vertical profiles above the aircraft limiting the number of data available below 3 km. The lidar measurement altitude range is of
the order of 6 km above the aircraft altitude with a 300 m vertical resolution and 2 minute temporal resolution corresponding a 10 km horizontal resolution. The system is described in Ancellet and Ravetta (1998), while performance during various airborne the applications is given in Ancellet and Ravetta (2003) where several comparisons with in situ measurements (ECC ozonesonde of airborne UV photometer) show no specific biases in clear air measurements. Measurements taken near clouds or thick aerosol layers are not included here since corrections of systematic errors related to aerosol interference are unreliable.
It corresponds to 20% of the lidar profiles recorded during the campaign.

The NASA-DC8 Ozone DIAL system and configuration implemented during the campaign is described by Richter et al. (1997). The instrument provides simultaneous zenith and nadir profiles to cover the troposphere and lower stratosphere. The measurement resolution for the archived data is 300 m in the vertical and approximately 70 km (3 min) in the horizontal. On all field experiments, the airborne DIAL $O_3$ measurements are compared with in situ $O_3$ measurements made on the
DC-8 during ascents, descents, and spirals and comparisons to ozonesondes during coincident overflights were conducted. In the troposphere, the DIAL $O_3$ measurements have been shown to be accurate to better than 10% or 2 ppb, whichever is larger (Browell et al., 1983, 1985). More recently, the precision of the DIAL $O_3$ measurements during the high-latitude SAGE-III Ozone Loss and Validation Experiment in 2003 (SOLVE II) were found to be better than 5% from near surface to




about 24 km and the accuracy was found to be better than 10% in comparison with Ny-Ålesund lidar, ozonesonde, and in situ DC-8 measurements (Lait et al., 2004). DIAL and Microwave Limb Sounder (MLS) $O_3$ measurements from the Polar Aura Validation Experiment (PAVE) in 2005 were found to agree within 7% across the 12-24 km altitude range (Froidevaux et al., 2008). Comparisons between DIAL and MLS were also examined in the upper troposphere and lower stratosphere

(215-100 hPa region) from data obtained during the International Intercontinental Chemical Transport Experiment (INTEX-B) field experiment, and these results show good agreement in lower stratosphere with decreasing performance of the MLS measurements into the troposphere (Livesey et al., 2008).

The ozonesonde monitoring was intensified in 2008 over North America in the framework of the ARCIONS initiative and the characteristics of the ozone measurements are fully described in Tarasick et al. (2010). Nearly daily sounding have been

made during the aircraft flight period at 4 stations over Canada in the latitude band between 53°N and 62°N. Only weekly sounding are made in the high latitude stations Alert and Resolute.

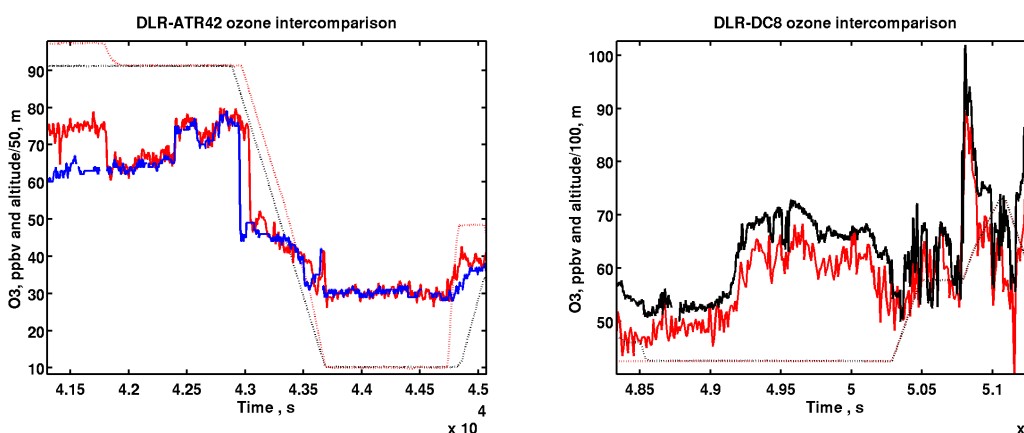

**Figure 2.** Ozone intercomparaison measurements in ppbv during two wing-tip to wing-tip flights over Greenland between the ATR-42 (blue) and the DLR-Falcon (red) on 14 July (left panel) and between the NASA-DC8 (black) and the DLR-Falcon (red) on 9 July (right panel). The aircraft altitude changes are also shown in m (multiply scale by 50 on right panel and by 100 on left panel)

The latitudinal cross sections of the ozone data set selected in this study are shown in Fig. 3 considering two domains to produce a bidimensional latitude-altitude plot over Canada (-160°W to -70°W) and Greenland (-70°W to -20°W). The horizontal distributions of the data set used for producing the latitude-altitude plots are shown in Fig. 1. For latitudes higher

than 80°N, the data comes from a limited number of observations: 8 sondes from Alert and 3 DC-8 flights from 8 July to 10 July. The locations of the ozonesonde stations are shown as red vertical bars in the latitudinal cross sections. In the troposphere at altitudes less than 8 km over Canada, lidar, in-situ and ozonesonde measurements contribute 67%, 20% and 13% of the ozone data set, respectively, while they correspond to 26% 69% and 5% respectively over Greenland (see Table 3 with the number of observations using each technique). The aircraft data (lidar and in-situ) therefore strongly contribute to the high

measurement density even though they correspond to a limited number of flying days, i.e. 18 days from 26 June to 18 July. In



**Table 1.** Characteristics of the ensemble of ozone measurements made with airborne instruments during summer 2008

| Instrument | Number of flights | Latitude range | Longitude range | Altitude range | Time period |
|---|---|---|---|---|---|
| ATR42 in-situ | 12 | 59°N-71°N | -60°W/-20°W | 0-7 km | 30/6-14/7 |
| ATR42 lidar | 12 | 59°N-71°N | -60°W/-20°W | 2-12 km | 30/6-14/7 |
| DC8 in-situ | 11 | 45°N-88°N | -132°W/-38°W | 0-12 km | 26/6-13/7 |
| DC8 lidar | 11 | 45°N-88°N | -132°W/-38°W | 0-15 km | 26/6-10/7 |
| F20 in-situ | 18 | 57°N-79°N | -65°W/-20°W | 0-11 km | 30/6-18/7 |

**Table 2.** Characteristics of the ECC sounding stations used during summer 2008

| Station | Number of ECC | Latitude | Longitude | Altitude range | Time period |
|---|---|---|---|---|---|
| Summit | 22 | 72.6°N | -38.5°W | 3.2-15 km | 6/6-22/7 |
| Alert | 8 | 82.5°N | -62.3°W | 0-15 km | 4/6-24/7 |
| Resolute | 8 | 74.7°N | -95°W | 0-15 km | 4/6-30/7 |
| Churchill | 16 | 58.7°N | -94°W | 0-15 km | 4/6-30/7 |
| Yellowknife | 19 | 62.5°N | -114.5°W | 0-15 km | 23/6-12/7 |
| Whitehorse | 15 | 60.7°N | -135.1°W | 0-15 km | 27/6-12/7 |
| Stonyplain | 16 | 53.55°N | -114.11°W | 0-15 km | 26/6-12/7 |

this work, ozone data are hourly averaged when they are in the same cell of a grid with a 0.5 x 0.5 degrees and a 1 km vertical resolution in order to avoid an oversampling of similar air masses. Only hourly averages are accounted for in Table 3.

For both regions similar ozone vertical distributions were observed with low ozone mixing ratio ≤ 40 ppbv below 3 km and similar upper troposphere lower stratosphere (UTLS) altitude range of 8-11 km between 60°N and 80°N (red and dark region with ozone mixing ratio > 150 ppbv). The average mixing ratio in the altitude range 4-8 km is of the order of 65 ppbv for both regions but the latitudinal gradients are more visible over Greenland than over Canada. Two mid-tropospheric ozone branches are seen in the latitude band 65°N-73°N and 78°N-85°N over Greenland, the first one being tilted to the South at 70°N and the second one to the North at 80°N.

## 3   WRF-Chem model simulation

### 3.1   Model description

For this study we use the regional Weather Research Forecasting model coupled with Chemistry (WRF-Chem) to study ozone during this period. WRF-Chem is a fully coupled, online meteorology and chemistry and transport mesoscale model. It has been



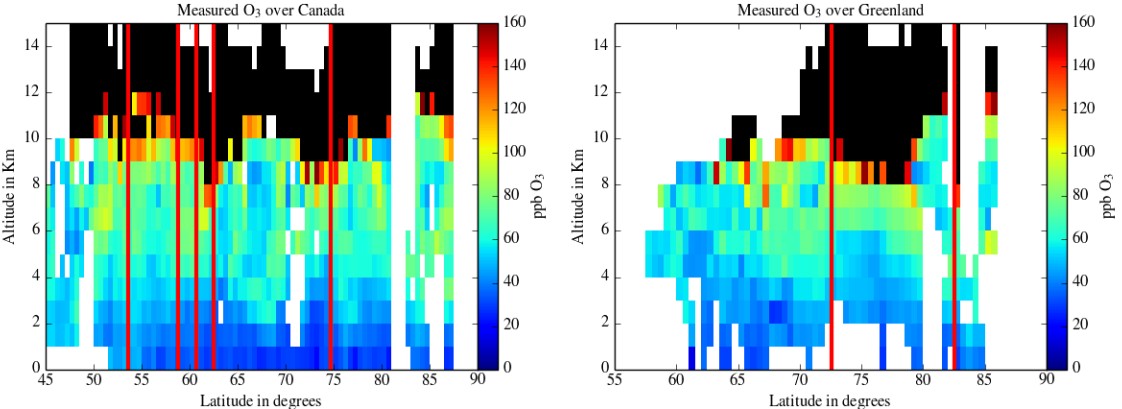

**Figure 3.** Latitudinal cross section of the measured ozone mixing ratio in ppbv over Canada (left panel) and over Greenland (right panel). The red bars show the location of ozonesonde stations. Black regions with $O_3 > 160$ ppbv correspond to the stratosphere.

successfully used in Arctic-focused studies in the past (Thomas et al., 2013; Marelle et al., 2015), for both gas phase and aerosol analysis. Initial meteorological conditions and boundaries are from the National Center for Environmental Prediction (NCEP) Global Forecast System (GFS) with nudging applied to temperature, wind and humidity every 6 hours. The simulation uses the Noah Land Surface model scheme with 4 soil layers, the YSU (Yonsei University, (Hong et al., 2006) planetary boundary

layer (PBL) scheme, coupled with the MM5 similarity surface layer physics, the Morrison 2-moment (Morrison et al., 2009) microphysics scheme, and the Grell-3D ensemble (Grell and Dévényi, 2002) convective implicit parametrization. The radiation schemes are the Goddard (Max and Suarez, 1994) and Rapid Radiative Transfer Model (Mlawer et al., 1997) for shortwave and longwave radiation respectively. Chemical boundary conditions were taken from the Model For Ozone and Related Chemical Tracers, version 4 (Emmons et al., 2010). For gas phase chemical calculations the CBM–Z (Zaveri and Peters, 1999) chemical

scheme is used and aerosols were calculated using the Model for Simulating Aerosol Interactions and Chemistry (Zaveri et al., 2008). The model was run from 15 March to 1 August 2008 using a polar stereographic grid ($100 \times 100$ km resolution) over a domain that covers most of the Northern Hemisphere, from about 28ºN. Vertically 50 hybrid layers up to 50 hPa are used with approximately 10 levels in the first two kilometers. The corresponding vertical resolution ranges from 100 m in the PBL to 500 m in the free troposphere. Anthropogenic emissions used are the ECLIPSE (Evaluating the CLimate and Air Quality ImPacts of

Short–livEd Pollutants) version 4.0 (Klimont et al., 2013), available in $0.5º \times 0.5º$ spatial resolution. Wildfire emissions were taken from GFED 3.1 (van der Werf et al., 2010), while aircraft/shipping emissions were from the RCP 6.0 scenario (Lee et al., 2009) and (Buhaug et al., 2009), respectively. Biogenic emissions were calculated online thanks to the Model of Emissions of Gases and Aerosols from Nature (Guenther et al., 2012). WRF-Chem also provides online dust and sea-salt emissions.

     Using observations from aircraft, surface stations and satellites, atmospheric model simulations of ozone have been evaluated

as part of POLMIP including WRF-Chem (Monks et al., 2015; AMAP, 2015). The model was run using different emissions and gas/aerosol schemes than in the POLMIP simulations, but the POLMIP results are still a good basis to choose WRF-



Chem. While all models have deficiencies in reproducing trace gas concentration in the Arctic, WRF-Chem performs better than many models in re-producing tropospheric ozone and CO, which are used here. Given the advantages of predicting also online stratosphere-troposphere exchange processes for this study, WRF-Chem is a good model for interpreting the ozone climatologies constructed from measurements in summer 2008.

## 3.2 Potential Vorticity calculations

The WRF-Chem model does not explicitly calculate potential vorticity (PV). As a result, for conducting the comparisons, PV was calculated offline based on WRF meteorological fields. Model potential temperature, total mass density, geopotential height and wind speed and direction was used to calculate PV per model gridbox.

For each model grid cell wind and temperature are interpolated from model vertical levels to the potential temperature in the center of the grid cell and the curl of the wind vector is calculated on the corresponding isentropic surfaces using the original model grid (100 km × 100 km) and the full model vertical resolution, i.e. approximately 500 m in the free troposphere. Potential vorticity is expressed in potential vorticity units (PVu) using the definition 1 PVu = $10^{-6}$ Kkg$^{-1}$m$^2$s$^{-1}$ . The main uncertainty in the PV calculation is related to the representation of smaller scales (50 km) than the model resolution (e. g. narrow stratospheric streamers near the tropopause).

## 3.3 Model results interpolations

The measurements used in this study vary in temporal and spatial resolution (section 2), where the model results are 3-hourly on a polar stereographic grid (described in section 3). In order to avoid favoring of data with the highest temporal resolution, the measured data was averaged to 1 min for all the in situ and lidar measurements. The ozone sondes are considered to be instantaneous at the time of the balloon release. A vertical resolution of 1km is used for the lidars and ozone sondes. The measured data is then gridded to 0.5º × 0.5º. If more than one measurements is available in the same grid box from the same campaign and instrument within a time frame of one hour, then these are considered to be the same measurement, and a mean value is calculated.

The model grid cell which is the most representative for the measurement's location in space is selected and a linear interpolation in time is done to calculate one minute increments from the 2 modelled values separated by 3 hours and surrounding the measurement time. This way the two resulting datasets (modelled and measured) both have a 1-min temporal resolution, and spatially only the vertical resolution is changed to set the altitude increment to 1 km. The model results, including PV, are also horizontally interpolated from the 100 km resolution to match the new 0.5º × 0.5º horizontal data resolution. Each model grid cell is split into 100 mini-grid cells. Then each mini-grid cell is assigned to the appropriate 0.5º × 0.5º final grid cell before calculating the new model values.





## 4 Comparison of measured and modeled ozone

The WRF-Chem ozone mixing ratios corresponding to the times and locations of the the June/July observations have been used to produce latitudinal cross sections comparable to the results shown in Fig. 3. The model ozone vertical cross sections for Canada and Greenland are shown in Fig. 4. The altitude and latitude ozone variability of the model is comparable with the observations over both regions. The vertical structure, i.e., the transition between the low ozone values below 3 km and the higher ozone concentration in the free troposphere and, the UTLS ozone variability, are well reproduced. The latitudinal gradient of the ozone concentration in the 4-8 km altitude range over Greenland is also visible in the simulation results where the two mid-tropospheric ozone branches are also visible at 70°N and 80°N. The agreement is less good for the latitudinal gradient over Canada where the low ozone values ($< 50$ ppbv) seen in the model at 65°N in the mid troposphere are not seen in the observations, although the signs of the latitudinal gradients seem correct.

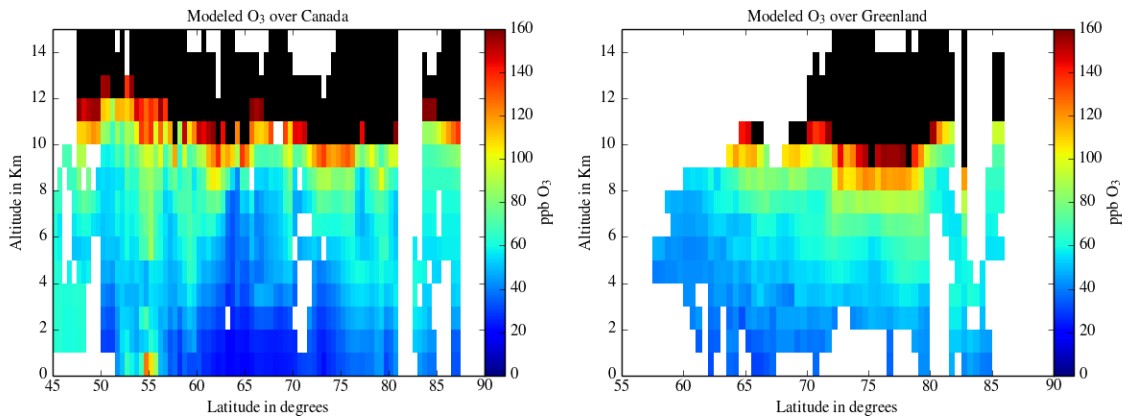

**Figure 4.** same as Fig. 3 for the WRF-Chem model ozone mixing ratio corresponding to the measurement sampling summarized in Tables 1 and 2.

To quantify the observation/model agreement the scatter plot of modeled versus measured ozone is also presented in Fig. 5 using a PV color scale to distinguish the tropospheric and stratospheric contributions. The correlation is of the order of 0.9 in the altitude range 0-15 km over both regions because the occurrence of stratospheric ozone intrusions are very well reproduced by WRF-Chem. The correlation is however between 0.5 and 0.7 in the troposphere only using observations with PV values less than 1 PVu. Considering that the ozone variability in the free troposphere is not very large ($< 50$ ppb), the spatial and temporal variability is still well reproduced by WRF-Chem even below the tropopause region. Even though the UTLS temporal variability is well reproduced by the model (Fig. 4), there is a significant underestimation of ozone by a factor 1.5 in the WRF-Chem simulation for the lowermost stratosphere (PV $>$2 PVu) (Fig. 5). In the troposphere there is also a negative bias of the model data of the order of -6 to -15 ppbv with the largest differences over Canada (see Table 5). A fraction of this tropospheric underestimate by WRF-Chem is likely related to the ozone underestimate in the lowermost stratosphere





which implies a smaller source of stratospheric ozone in the free troposphere. This likely originates from the ozone climatology used to initialize the model in the stratosphere. The other part of this 10-ppbv bias is due an uncertainty of the lightning $NO_x$ contribution in the WRF-Chem simulation and/or an underestimate of the vertical transport of continental emissions to the mid- and upper troposphere. Wespes et al. (2012) have shown that, the lightning $NO_x$ ozone source and vertical transport of continental emissions both contribute to 15% of the ozone concentration in the free troposphere at latitudes higher than 60°N in summer. Despite these flaws in the WRF-Chem simulations the spatial ozone variability is still very valuable because it compares rather well with the observations. The PV and CO distribution available from the WRF-Chem simulations can be used to examine the respective roles of photochemistry and STE on the observed ozone distribution.

## 5    Latitudinal ozone distribution at high latitudes

In this section, we investigate the mean latitudinal and vertical ozone gradient in the troposphere using the POLARCAT observations. To quantify these gradients, the ozone cross sections plotted in Fig.3 have been divided into several regions using the model CO latitudinal cross sections. The vertical boundaries are defined according to the mean ozone vertical profile in the troposphere, i.e. the the depth of the low ozone values layer in the lower troposphere and the downward extent of the tropopause region. The variability of the CO distribution is only taken from the model because CO is not available for many of the ozone observations (ozonesonde and lidar profiles).

### 5.1    Measurements over Canada

For the data taken over Canada, 5 regions have been considered to calculate the mean $O_3$ latitudinal gradient. They correspond to the blue boxes with the label 1, 2, 3, 4 and 8 shown in the CO latitudinal distribution plotted in Fig. 6. The regional extent of the zones have the same order of magnitude to make them comparable. Two are below 4 km in the altitude range where the lowest ozone values were recorded for latitudes > 60°N. The boundary between these two boxes is set according to the strong CO latitudinal difference due to the biomass burning emissions south of 65°N and lack of local emissions of ozone precursors in the region between 65°N and 80°N. In the altitude range 4-8 km, which corresponds to the largest tropospheric ozone values, two other regions were defined. The latitude boundary at 63°N is again prescribed according to the latitude where CO and $O_3$ concentrations are simultaneously decreasing. The last box corresponds to the tropospheric observations at high latitudes (> 80 °N) where CO concentrations are increasing especially in the altitude range 3-8 km. These ozone observations were mainly made in northeastern Canada by the DC8 aircraft. The CO distribution derived from the WRF-Chem simulation is consistent with the analysis of CO observations by Bian et al. (2013), who show the major influence of boreal and Asian emissions on the ARCTAS-B CO observations.

The mean and median measured $O_3$ mixing ratio for the different boxes are shown in Table 3 including the number of observations from the different measurement techniques (lidar, in-situ and sondes). The mean and median $O_3$ and CO mixing ratios from the model are also reported in Table 4 and the statistics relevant to the model evaluation are reported in Table 5. The -10 to -15 ppbv differences between the model and measured ozone in zones 3, 4 and 8 above 4 km are consistent with the bias



of the model in the troposphere over Canada discussed in the previous section. This bias is small (0 to -4 ppbv) in the lower troposphere, showing that the emissions used in the model simulation are good enough to calculate the $O_3$ photochemical production.

The negative latitudinal gradient of ozone between zones 3 and 4 ($\Delta O_3$= -6 ppbv for the measurements and $\Delta O_3 \approx$ -8 ppbv for the model) and to a lesser extent between zones 1 and 2 ($\Delta O_3$= -4 ppbv for the measurements and $\Delta O_3$= -5 ppbv for the model) are correlated with a significant negative latitudinal CO gradient. For latitudes lower than 65°N there is a strong standard deviation because some of the measurements were taken very close to fresh biomass burning sources. Sampling of the biomass burning sources during ARCTAS by the DC8 aircraft between 50°N and 63°N has been already discussed in several papers (Singh et al., 2010; Alvarado et al., 2010; Thomas et al., 2013). In the mid-troposphere the CO enhancement in zone 3 where the largest $O_3$ mixing ratio (70 ppbv) is recorded, is 130 ppbv, well above the CO tropospheric baseline of 60 ppbv (the strong difference between the mean and the median corresponds to the sampling of one biomass burning plume with CO > 500 ppbv which biases the mean). Zone 8 data in Tables 3 and 4 includes the aircraft sampling over both Canada and Greenland for the high latitude boxes (> 80°N) because they characterized the same regions with similar ozone and CO distribution. The ozone mean for zone 8 (70 ppbv) is similar to the ozone mean found in zone 3 while CO is also increasing again at high latitudes (40 ppbv above the CO tropospheric baseline of 60 ppbv).

The average PV latitudinal cross section is also calculated for the ozone dataset as it is a good tracer of the latitudinal variability of the stratospheric $O_3$ source (Fig. 7). Although frequent stratospheric air mass intrusions are seen in the 8-10 km altitude range at latitudes higher than 55°N, the role of the stratospheric source is not clearly visible in the average PV values in the free troposphere. Mean PV values larger than 0.5 PVu are only seen for latitudes higher than 75°N in the 3-6 km altitude range (the large PV values at altitudes below 2 km should not be considered as they are related to the low level cyclonic circulation due the orographic circulation around the Greenland ice cap). This suggests that the latitudinal ozone gradients between zones 3 and 4 is not related to the latitudinal distribution of stratospheric intrusion. Ozone mixing ratios larger than 70 ppbv (Fig. 3) are however correlated with the descending high PV tongue at latitudes between 77°-87°N (Fig. 7). Because the analysis of the ozone spatial distribution shows contrasted behaviors of the ozone-PV relationship, we also used the PV versus $O_3$ mixing ratio scatter plot for all the tropospheric ozone data, which shows a poor Pearson correlation (r< 0.3) between ozone and stratospheric intrusion (Fig. 7). Using the color scale of the scatter plot to separate the tropospheric air masses and the UTLS region in the 8-10 km altitude range, one can see that a significant number of high tropospheric ozone mixing ratios (> 70 ppbv) are indeed related to PV less than 1 PVu. The difference in the slopes of the regression line, when including or not the PV values larger than 1 PVu (red versus green lines in Fig. 7), is also larger than similar variation observed at mid-latitudes in Europe where the $O_3$ to PV ratio decreases from 30 to 150 ppbv/PVu (Ravetta et al., 1999). This suggests that the ozone variability is not from stratospheric air that mixed into the free troposphere during summer 2008, instead the ozone cycle was mainly driven by emissions and photochemistry over Canada.





**Figure 5.** Linear (left) and log scale (right) scatter plot of WRF-Chem modeled versus measured $O_3$ mixing ratio in ppbv for Canada (top panel) and Greenland (bottom panel) domains. The black lines are the one to one line and the PV color scale distinguishes stratospheric (red and black dots) and tropospheric data (blue and green dots). The regression line parameters and the Pearson correlation coefficient with its p-value are coloured in red and green for the PV range 0-4 PVu and 0-1 PVu, respectively.





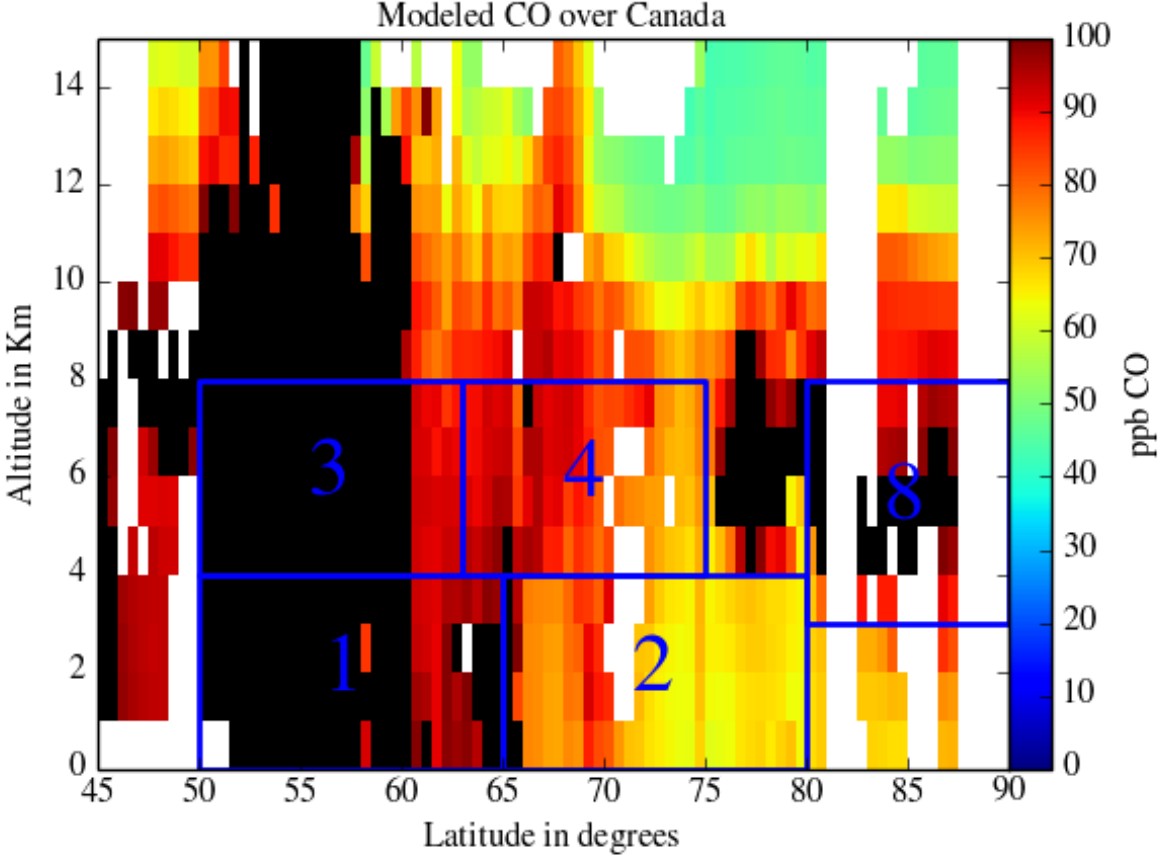

**Figure 6.** Latitudinal cross section of the tropospheric CO profiles over Canada for the measurement spatio-temporal distribution from the June/July 2008 WRF-Chem simulation. The blue boxes correspond to the regions where data are averaged for discussing vertical and latitudinal gradients in Table 4.





**Table 3.** Mean, median and standard deviation of the observed ozone mixing ratio in the different boxes shown in Fig. 6 and 8, except for the two boxes at latitudes $> 80^{\circ}$N which have been merged in the last row of the table.

| Zone N° | Latitude range | Longitude range | Altitude km | $O_3$ Mean ppbv | $O_3$ Median ppbv | $O_3$ St.Dev. ppbv | Number lidar | Number in-situ | Number ECC |
|---|---|---|---|---|---|---|---|---|---|
| 1 | 50°/65°N | -132°/-70°W | 0-4 | 45.8 | 45.3 | 10.8 | 1046 | 422 | 243 |
| 2 | 65°/80°N | -132°/-70°W | 0-4 | 42.3 | 40.9 | 11.4 | 401 | 27 | 28 |
| 3 | 50°/63°N | -132°/-70°W | 4-8 | 69.4 | 68.0 | 18.8 | 1184 | 349 | 240 |
| 4 | 63°/75°N | -132°/-70°W | 4-8 | 63.4 | 62.0 | 16.5 | 360 | 123 | 28 |
| 5 | 60°/80°N | -70°/-20°W | 0-4 | 45.1 | 45.4 | 8.0 | 243 | 603 | 21 |
| 6 | 57°/65°N | -70°/-20°W | 4-8 | 57.4 | 56.1 | 13.8 | 22 | 203 | 0 |
| 7 | 65°/75°N | -70°/-20°W | 4-8 | 68.7 | 68.1 | 16.3 | 320 | 906 | 81 |
| 8 | 80°/87°N | -132°/-20°W | 3-8 | 69.8 | 67.6 | 14.8 | 166 | 48 | 35 |

**Table 4.** same as Table 3 for the WRF-Chem model $O_3$, CO mixing ratio and PV $75^{th}$ percentile.

| Zone N° | Latitude range | Longitude range | Altitude km | $O_3$ Mean ppbv | $O_3$ Median ppbv | CO Mean ppbv | CO Median ppbv | PV 75th PVu |
|---|---|---|---|---|---|---|---|---|
| 1 | 50°/65°N | -132°/-70°W | 0-4 | 45.9 | 42.9 | 441.5 | 129.6 | - |
| 2 | 65°/80°N | -132°/-70°W | 0-4 | 38.3 | 37.7 | 71.8 | 67.8 | - |
| 3 | 50°/63°N | -132°/-70°W | 4-8 | 60.4 | 55.4 | 509.7 | 129.0 | 0.35 |
| 4 | 63°/75°N | -132°/-70°W | 4.8 | 45.7 | 47.9 | 87.8 | 87.6 | 0.31 |
| 5 | 60°/80°N | -70°/-20°W | 0-4 | 45.5 | 45.6 | 66.4 | 66.1 | - |
| 6 | 57°/65°N | -70°/-20°W | 4-8 | 51.4 | 49.0 | 83.9 | 78.8 | 0.45 |
| 7 | 65°/75°N | -70°/-20°W | 4-8 | 62.8 | 61.1 | 76.2 | 75.8 | 0.44 |
| 8 | 80°/87°N | -132°/-20°W | 3-8 | 57.7 | 58.0 | 100.0 | 98.7 | 0.53 |





**Table 5.** Metrics of the WRF-Chem $O_3$ simulation performance for the regions reported in Table 4: mean bias, root mean square error (RMSE), normalized mean bias

| Zone N° | Latitude range | Longitude range | Altitude km | Mean bias ppbv | RMSE ppbv | Normalized mean bias,% |
|---|---|---|---|---|---|---|
| 1 | 50°/65°N | -132°/-70°W | 0-4 | 0.1 | 24.0 | 0.3 |
| 2 | 65°/80°N | -132°/-70°W | 0-4 | -4 | 10.3 | -9.5 |
| 3 | 50°/63°N | -132°/-70°W | 4-8 | -9 | 24.9 | -12.9 |
| 4 | 63°/75°N | -132°/-70°W | 4.8 | -15 | 21.9 | -28.0 |
| 5 | 60°/80°N | -70°/-20°W | 0-4 | 0.4 | 7.5 | 0.8 |
| 6 | 57°/65°N | -70°/-20°W | 4-8 | -6 | 15.3 | -10.6 |
| 7 | 65°/75°N | -70°/-20°W | 4-8 | -6 | 17.1 | -8.6 |
| 8 | 80°/87°N | -132°/-20°W | 3-8 | -13 | 18.0 | -17.2 |





**Figure 7.** Latitudinal cross section of the average PV profiles over Canada for the summer measurement distribution using the WRF-Chem simulation (top panel) Linear (left) and log scale (right) scatterplot of measured $O_3$ in ppbv versus PV in PVu using an altitude color coded scale (bottom panel). Stratospheric data points with PV > 4 PVu are not included. The regression line parameters and the Pearson correlation coefficient with its p-value are also given for the PV range 0-4 PVu (red) and 0-1 PVu (green).





## 5.2 Measurements over Greenland

The same procedure was applied to the latitudinal distribution over Greenland. The regions considered for the latitudinal gradient analysis are slightly different because the CO latitudinal cross sections and the ozone vertical structure are different. The selected regions are shown by the blue boxes in Fig. 8. Only one zone is chosen between 60°N and 80°N for the altitude range below 4 km where there are the lowest tropospheric $O_3$ and CO concentrations values. The CO latitudinal gradient is indeed very weak in the lower troposphere between 60°N and 80°N ($< 5$ ppbv). However, like over Canada, two boxes are chosen in the altitude range 4-8 km because of their differences in term of CO and $O_3$ concentrations. An additional box corresponds to the tropospheric observations at high latitudes ($> 80$ °N) where the CO concentrations are also higher especially in the altitude range 3-8 km. This is related to the observations mainly made in northwestern Greenland by the DC8 aircraft.

Table 5 shows that, there is no $O_3$ underestimate by the model in the lower troposphere and the model bias above 4 km is of the order of -6 ppbv. This suggests that the model performance over Greenland, away from the continental sources, is better than over Canada. The remaining -6 ppbv bias can be easily explained by the stratospheric $O_3$ climatology having too low values, while ligthning is known to be less important over Greenland (Cecil et al., 2014; Christian et al., 2003). The CO concentration in the lower troposphere (zone 5 of Table 3) is close to the tropospheric baseline, while $O_3$ is not markedly different from the values seen over Canada. The positive latitudinal $O_3$ gradient between zones 6 and 7 in the mid-troposphere above 4 km ($\Delta O_3$=12 ppbv) is two times larger than the latitudinal gradient over Canada, but the $O_3$ mid-tropospheric concentration over Greenland is not significantly higher than its counterpart over Canada (62 ppbv versus 65 ppbv). The corresponding negative CO latitudinal gradient between zone 6 and 7 is weak (difference of the order of 8 ppbv), but it is however anticorrelated with the ozone gradient. The CO excess above baseline is only 20 ppbv in zone 6, where CO is maximum. The anticorrelation between the $O_3$ and CO latitudinal gradient may correspond to more occurrences of stratospheric ozone intrusion in zone 7 at latitudes higher than 65°N during the POLARCAT period. Less CO due to pristine air from the Arctic lower troposphere would lead to smaller $O_3$ concentrations according to Fig. 3.

Looking at the average PV distribution extracted from the WRF-Chem simulation (Fig. 9), frequent stratospheric air mass intrusions are seen in the altitude range 8-10 km especially at 75°N (PV $> 1$ PVu). PV values are also larger in the mid troposphere over Greenland than over Canada (75th PV percentile is 0.45 PVu over Greenland but 0.3 PVu over Canada). The PV versus $O_3$ mixing ratio scatter plot also shows a higher Pearson correlation (r$> 0.4$) between ozone and stratospheric intrusion than over Canada (Fig. 9). Unlike the results obtained over Canada, the difference between $O_3$ to PV ratio including or not PV values larger than 1 (red versus green regression line) is smaller (200 ppbv/PVu instead of 400 ppbv/PVu for Canada)). It is consistent with a larger fraction of ozone from the UTLS over Greenland. However there is not a significant latitudinal PV gradient in the mid-troposphere between zones 6 and 7 where a positive 12 ppbv ozone latitudinal gradient is detected. At high latitudes above 80°N, the PV and CO latitudinal distribution in the troposphere are almost identical to the latitudinal cross section over Canada (Fig. 7), which support merging data in a single zone 8 for the statistics reported in Table 3 and 4.





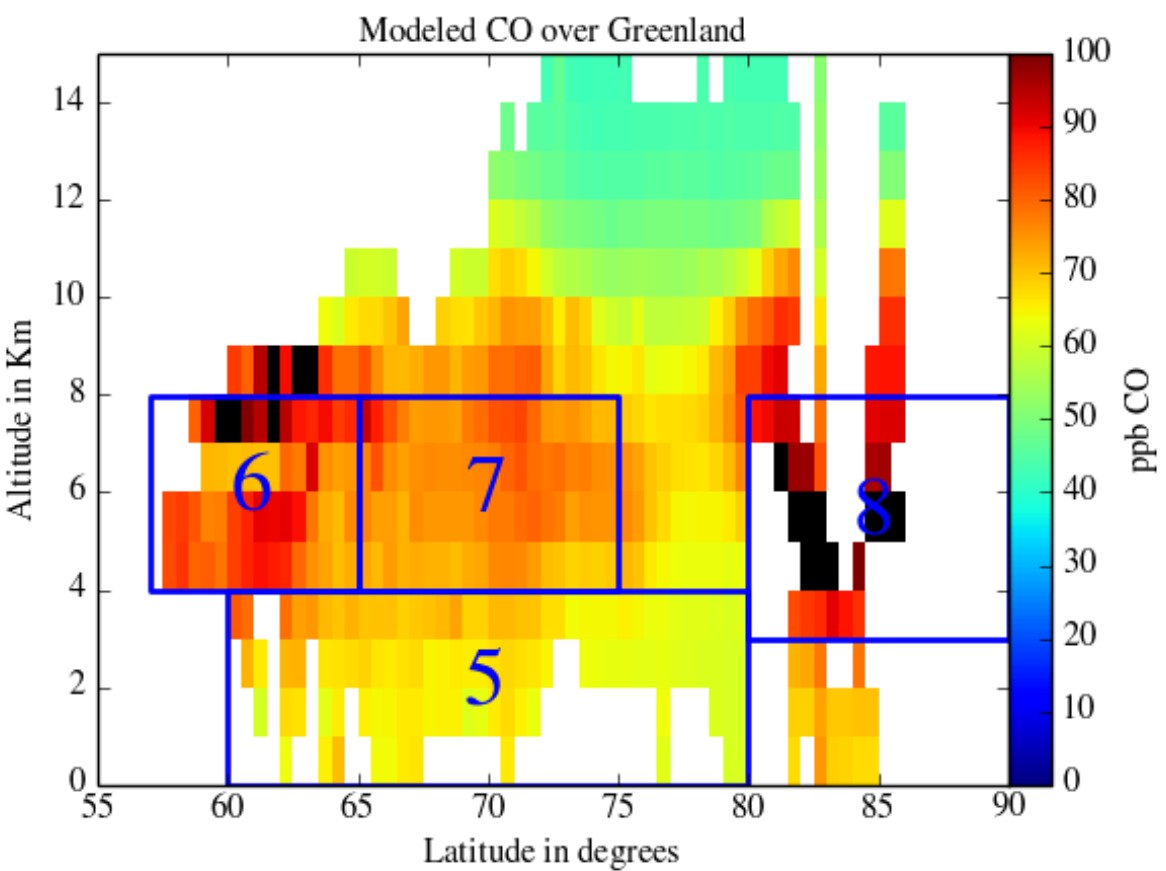

**Figure 8.** same as Fig. 6 for the Greenland region.

.





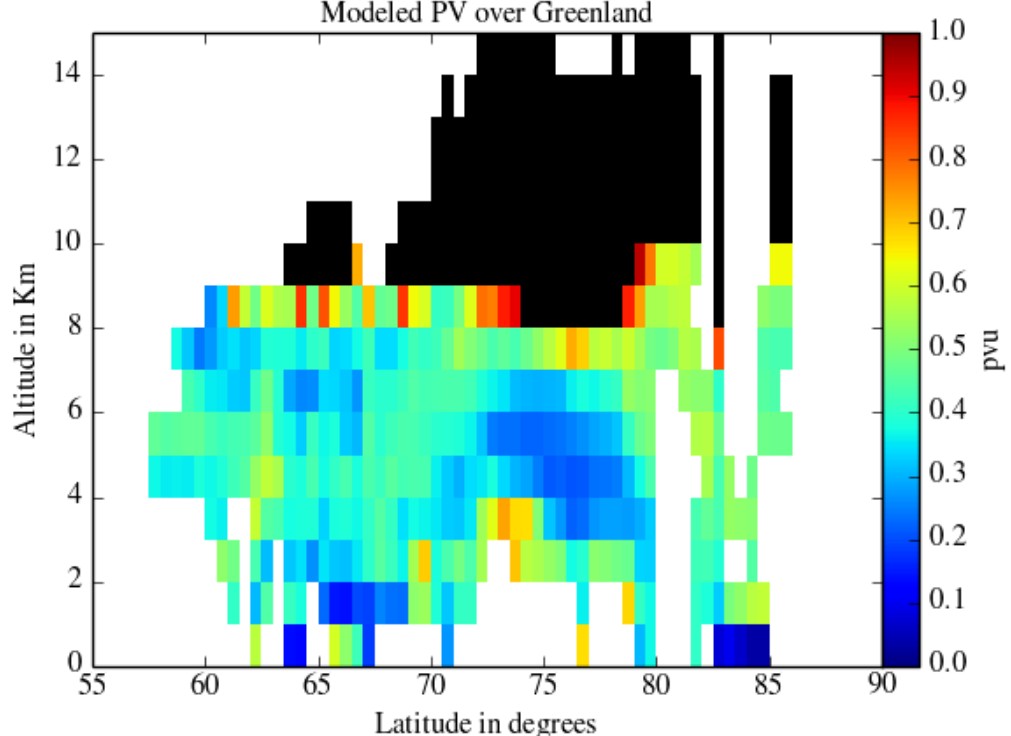

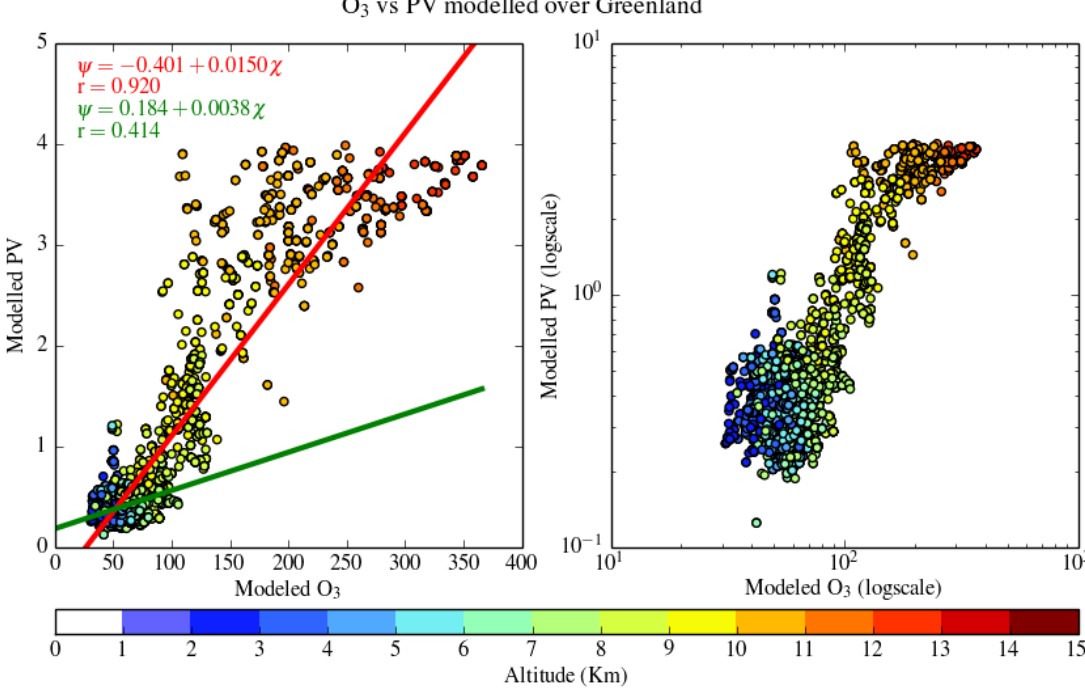

**Figure 9.** same as Fig. 7 for the Greenland region.





## 5.3 Discussion

In the lower troposphere (0-4 km), the fact that ozone is lower than 50 ppbv everywhere (except close to the fires south west of Hudson Bay) is expected considering a near zero net ozone photochemical production (+0.3 ppbv/day near the surface and -0.3 ppbv/day in the lowermost free troposphere) and a weak influence of STE below 5 km in Northern Canada (Walker et al.,
2012; Mauzerall et al., 1996).

In the free troposphere above 4 km, the negative CO latitudinal gradient in the latitude range 50°N to 75°N is seen over both Greenland and Canada. The large CO latitudinal gradient over Canada can only be explained when considering the biomass burning plume at the continental scale which is superimposed onto the anthropogenic emissions south of 50°N. The position of the biomass burning plume is shown by the MODIS monthly mean aerosol optical depth (last panel of Fig. 11) with maxima
both south west of Hudson Bay and over the Atlantic ocean in the latitude band 50°N to 60°N. The ozone latitudinal gradient is also negative and correlated with CO over Canada, which can be explained by the ozone photochemical production in the Canadian biomass burning plumes. Thomas et al. (2013) indeed showed a 3 ppbv ozone increase downwind of biomass burning emissions and an enhancement ratio $\Delta O_3/ \Delta CO$ ranging from 0.1 near the fires to 0.5 downwind. It is consistent with a $\Delta O_3$ of 6-8 ppbv between zone 3 and 4 over Canada where a 40 ppbv difference in the median CO value is found in the model
simulation. The contribution of other emissions sources can be also estimated using 4-day backward trajectories calculated with the FLEXTRA model and T213/L91 ECMWF analysis (Fig. 10). The differences in the long range transport between zones 3 and 4 over Canada are not very large with a similar inflow from northern Pacific according to the upper panels of Fig. 10. Regional variability of STE has also a weak influence during this period considering the weak dependency of $O_3$ with PV over Canada. Local emissions from biomass burning are the most reasonable explanation for the ozone latitudinal gradient
between 50°N and 75°N.

Over southern Greenland the effect of ozone photochemical production related to fire plumes and the North American anthropogenic emissions apperas to be less than 4 ppbv, considering the weak CO gradient between zones 6 and 7 ($\approx$-8 ppbv). The negative $O_3$ gradient due to increased photochemical production in zone 6 can be then easily counterbalanced by another source in zone 7 to explain the positive latitudinal $O_3$ gradient over Greenland. STE ozone source may contribute because of
the better correlation between $O_3$ and PV over Greenland than Canada and larger PV over Greenland. Frequent tropopause polar vortices developing over Baffin Bay and Davis Strait still occur during the summer period (Cavallo and Hakim, 2010). However the absence of a clear PV latitudinal gradient between 55°N and 75°N in the model simulation cannot explain the significant +12 ppbv ozone latitudinal gradient. Therefore STE certainly contributes in the ozone level over both southern and northern Greenland, but it does not explain the largest $O_3$ values found in zone 7. Looking at the long range transport plot
for zones 6 and 7 (lower panels of Fig. 10), multiple mid-latitude sources including North America , Europe and even cross Arctic transport can lead to a more efficient $O_3$ photochemical production in zone 7 compared to zone 6. Wespes et al. (2012) concluded using the Model MOZART-4 that anthropogenic pollution from Europe dominates $O_3$ concentrations in summer 2008 in the Arctic, while Roiger et al. (2011) shows using aircraft measurements that Asian anthropogenic pollution can be mixed with stratospheric air masses over Greenland in the 75°-80°latitude band.





For the free troposphere at high latitudes above 80 °N, STE contribution is maximum in this region for the 4-8 km altitude range according to the average PV distribution shown in Fig.7 or Fig.9. The 75th percentile of the PV distribution is also > 0.5 PVu in zone 8 while it is 0.3 PVu at latitudes lower than 75°over Canada and 0.4 PVu over Greenland. Zängl and Hoinka (2001) discussed horizontal gradient of the tropopause height over the Arctic using ECMWF analysis and radiosondes. The region with the largest horizontal gradient is displaced to the North near 80 °N in July over North America, while vertical tilting of the isentropic surfaces remains small. These conditions are favorable for isentropic motion across the tropopause with more efficient STE. However CO is also increasing again at high latitude because of the transport Asian and North Siberian pollution sources according to the air mass transport pathway for zone 8 (Fig. 11). Such a mixture of stratospheric ozone and Asian pollution has been already suggested by Roiger et al. (2011) to explain the ozone concentrations observed at very high latitudes in the Arctic. This study using more ozone measurements leads to the same conclusions.

## 6  Conclusions

The purpose of this work is to provide a complete picture of Arctic ozone using measurements available during the summer 2008 POLARCAT campaigns over Canada and Greenland where 3 aircraft were deployed and 7 ozonesonde stations intensified their ozone vertical profiling. This is the first case of such complete temporal and geographical coverage, specifically we take advantage of the large number of airborne lidar profiles representing 67% of the measurements over Canada and 26% over Greenland. A good correspondence of the measured $O_3$ vertical and latitudinal distribution is found with model results from WRF-Chem, especially over Greenland. A negative $O_3$ bias of -6 to -15 ppbv between the model and the observations in the free troposphere over 4 km, especially over Canada, is partly related to the model deficiencies in the lowermost stratosphere, because of the ozone climatology used for stratospheric ozone. The measured ozone climatology establihed in this paper can serve as a benchmark for future model evaluation at high latitudes.

The analysis of the WRF-Chem model simulation is also useful to discuss the relative influence of tropospheric ozone sources at high latitude in summer. Ozone average concentrations are of the order of 65 ppbv at altitudes above 4 km both over Canada and Greenland, while they are less than 50 ppbv in the lower troposphere. For Canada, the analysis of the model CO distribution and the weak correlation (< 30%) of $O_3$ and PV suggest that stratosphere-troposphere exchange (STE) is not the major contribution to tropospheric ozone at latitudes less than 70°N, because local biomass burning (BB) emissions were significant during the 2008 summer period. Conversely over Greenland, significant STE is found according to the better $O_3$ versus PV correlation (> 40%) and the higher 75th PV percentile.

A weak negative latitudinal summer ozone gradient -6 to -8 ppbv is found over Canada in the mid troposphere between 4 and 8 km because the $O_3$ photochemical production due to the BB emissions mainly takes place at latitudes less than 65°N, while STE plays only a significant role at latitude higher than 70°N. A positive ozone latitudinal gradient of 12 ppbv is observed in the same altitude range over Greenland not because of an increasing latitudinal influence of STE, but because of different long range transport from multiple mid-latitude sources (North America, Europe and even Asia for latitudes higher than 77°N).



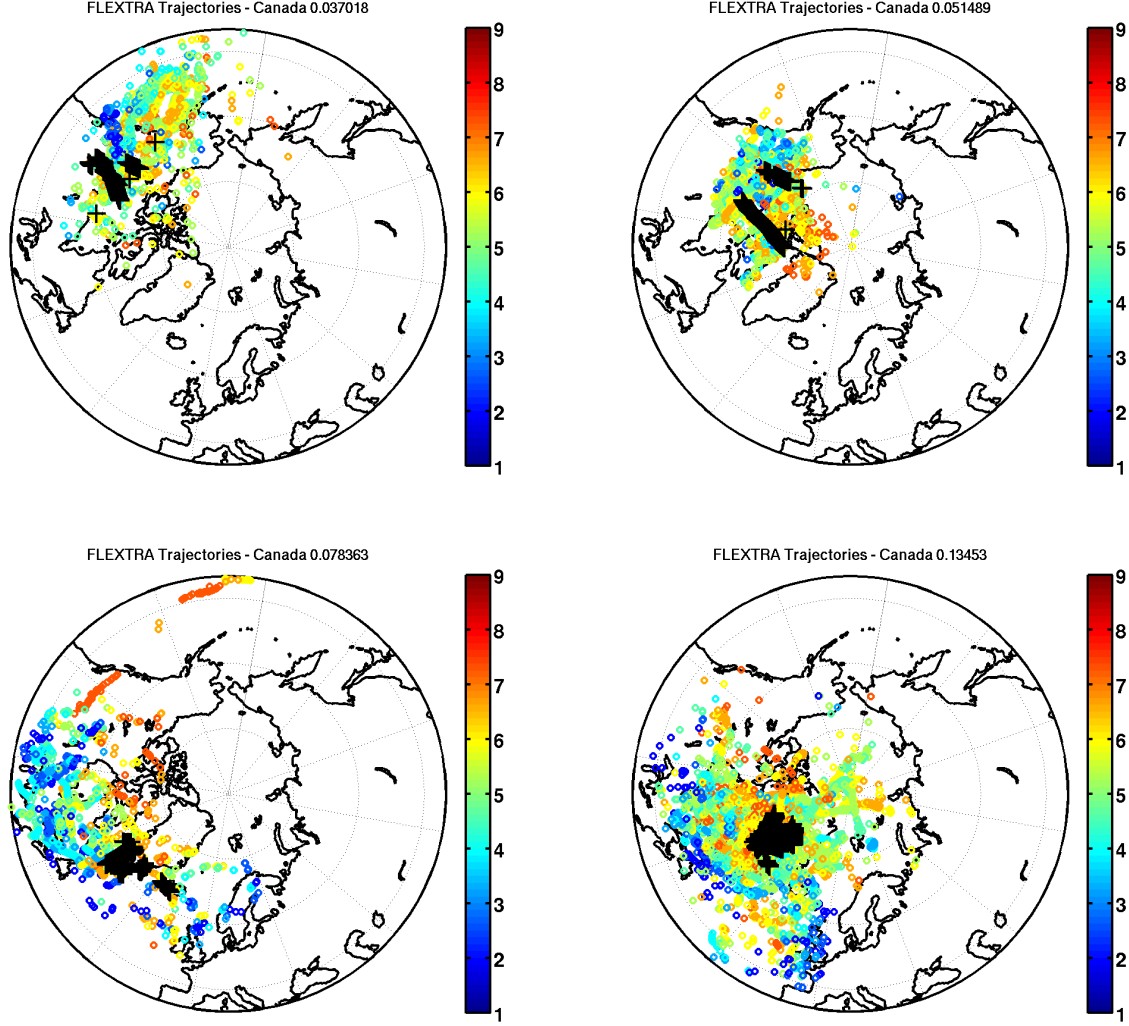

**Figure 10.** Map of the air mass daily positions using 4-days FLEXTRA trajectories which correspond to the observations in zones 3 and 4 over Canada (top row) and in zones 6 and 7 over Greenland (bottom row). Black crosses show the measurement positions and the color scale is the air mass altitude in km. The fraction is the relative number of trajectory positions reaching the tropopause (PV=1.5 PVu).

*Acknowledgements.* We are very grateful to the support of the Meteo France/CNRS/CNES UMS SAFIRE for the ATR-42 aircraft deployment over Greenland. This work was supported by funding from ANR and LEFE INSU/CNRS (CLIMSLIP project) and from the ICE-ARC programme from the European Union 7th Framework Programme, grant number 603887, ICE-ARC contribution number XXXXXX . The FLEXTRA team (A. Stohl, and co-workers) is acknowledged for providing the FLEXTRA code. NASA and DLR are acknowledged for their support to the deployment of the DC8 and Falcon 20 aircraft. WOUDC is acknowledged for providing the ozonesonde data.





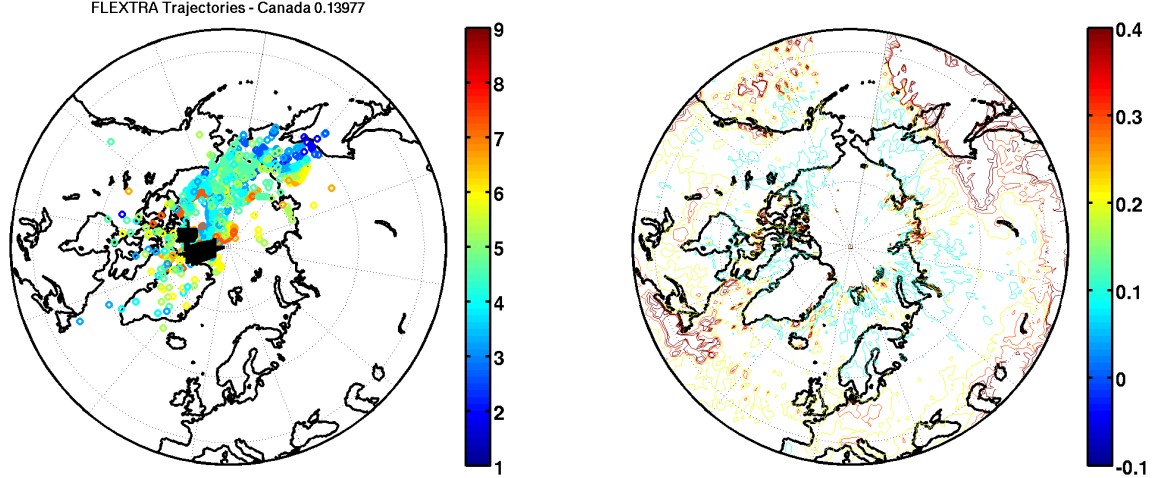

**Figure 11.** same as Fig.10 for the observations at latitudes $> 80^\circ$N in zone 8 (left). Map of MODIS aerosol optical depth at 0.55 $\mu$m from 15 June to 15 July 2008 (right).

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
