# Peer review of "Analysis of the latitudinal variability of tropospheric ozone in the Arctic using the large number of aircraft and ozonesonde observations in early summer 2008"

_Atmospheric Chemistry and Physics, 2016_

## Referee Comment (RC1) · Anonymous Referee #1 · 13 Jun 2016

The paper meets the AMT criteria and can be published after minor revision and technical corrections. An important feature of this paper is that it is not clear if only nighttime tropospheric O3 measurements are performed. The authors have to clarify this everywhere needed in the manuscript (i.e. in Fig. 17, it is not mentioned; and also in all Figs., this should be mentioned).

The annotated paper is attached to this report.

Please also note the supplement to this comment:

[Figure]

http://www.atmos-chem-phys-discuss.net/acp-2016-422/acp-2016-422-RC1-supplement.pdf

[Figure]

**Supplement:**

[Figure]

**Analysis of the latitudinal variability of tropospheric ozone in the Arctic using the large number of aircraft and ozonesonde observations in early summer 2008**

Gerard Ancellet[1], Nikos Daskalakis[1], Jean Christophe Raut[1], Boris Quennehen[1], François Ravetta[1], Jonathan Hair[2], David Tarasick[3], Hans Schlager[4], Andrew J. Weinheimer[5], Anne M. Thompson[6], Bryan Johnson[7], Jennie L. Thomas[1], and Katharine S. Law[1]

[1]LATMOS/IPSL, UPMC Univ. Paris 06 Sorbonne Universités, UVSQ, CNRS, Paris, France
[2]NASA Langley Reasearch Center, Hampton, VA, USA
[3]Environment and Climate Change Canada, Downsview, ON, Canada
[4]Institut für Physik der Atmosphäre, DLR, Oberpfaffenhofen, Germany
[5]NCAR, Boulder, CO, USA
[6]NASA/GSFC, Greenbelt, MD, USA
[7]NOAA/Earth System Research Laboratory (ESRL), Boulder, CO, USA

Correspondence to: Gerard Ancellet : gerard.ancellet@upmc.fr

*(handwritten annotations:)* put the goals in the form presented in page 3, lines 6-10.

*This abstract has to be rephrased. Do not start with the goals of this paper. Start by: "During the International Polar Year...". After, you can...*

→ See General comments

[revised manuscript text omitted]

---

## Referee Comment (RC2) · Anonymous Referee #2 · 9 Aug 2016

Review of "Analysis of the latitudinal variability of tropospheric ozone in the Arctic using the large number of aircraft and ozonesonde observations in early summer 2008" by Ancellet et al.

General comments:

The paper provides a detailed description of tropospheric ozone measurements during the POLARCAT campaign in summer 2008, and a series of analysis of its variability and possible sources affecting the variability by using WRF-Chem chemistry-transport model. The description is intensive and the analysis is generally sound, reaching some

interesting conclusions. I found the paper be a nice piece of work contributing to better understanding of sources and processes affecting the summertime ozone at high-latitudes. The paper is well within the scope of ACP, and can eventually be published. However, I found some descriptions/illustrations are redundant. There are many errors in English. Some figures look preliminary. These made me difficult to follow what the authors are trying to tell us. So, I would suggest some technical suggestions that should be addressed before publication.

Specific comments:

Figures 5, 7, and 9: The same data are plotted in both linear and logarithmic scales. I doubt if the authors really need logarithmic plots, as they do not discuss much on the log plots. The log plots seem redundant to me.

English and technical errors:

There are many errors in English, for example in Abstract:

L1: The goals of the paper are ...

L10: The average ozone concentrations are 65 ppbv ...

L14: ... modeled CO ...

L18: ... ozone gradient of -6 to -8 ppbv . . .

Abstract, L7: Ozone, CO and . . . is too much detail in Abstract, and can be removed.

P7, L17: MEGAN (Model of ...)

Figure 2 caption: Intercomparison of ozone measurements ...

Captions in other figures: Measured O3 (ppbv), Modeled O3 (ppbv) - need units!

Figures 3 and 4:

I would make difference plots between observation and model, to illustrate where in

height and latitude the model is good or bad. This is not necessary but please consider.

---

## Author Comment (AC1) · 23 Sep 2016

We thank reviewer 1 for his careful review and the useful annotated paper. We apologize for the large number of English errors and the new manuscript has been carefully edited by one of the English speaking co-author. Find below the answers to the questions raised by the reviewer and the revised paper is provided as a supplement document

Reviewer:"The paper meets the ACP criteria and can be published after minor revision and technical corrections. An important feature of this paper is that it is not clear if only

nighttime tropospheric O3 measurements are performed. The authors have to clarify this everywhere needed in the manuscript (i.e. in Fig. 17, it is not mentioned; and also in all Figs., this should be mentioned)".

There is no figure 17 but we specify now that the ozone lidar performances are given for daytime mesurements in section 2. It is true for both the ALTO lidar and the NASA airborne lidar. Regarding the questions about daytime versus nighttime lidar measurements. All the aircraft flights are during the day and the ozone measurement performances are given for daytime measurement. It is said more clearly in section 2.

The annotated paper is attached to this report. The main remarks in the annotated are: Reviewer "Need to rephrase line 18-20 p. 2 and line 6-7 p. 4"

The two new pieces of text are now:

" The ozone distributions over North America have been discussed for the spring period at high latitude using the Tropospheric Ozone Production about the Spring Equinox (TOPSE) and Arctic Research of the Composition of the Troposphere from Aircraft and Satellites (ARCTAS) data set in several publications (Browell 2003, Wang 2003, Olson 2012, Koo 2012) showing: (i) frequent occurrence of ozone depletion events (ODE) in the planetary boundary layer (PBL), (ii) a net O3 photochemical production rate equal to zero throughout most of the troposphere, (iii) a latitudinal increase of the tropospheric ozone concentrations due to transport from mid-latitudes and Stratosphere-Troposphere Exchange (STE)"

"The system is fully described in Ancellet and Ravetta (1998), and the instrument performances for different examples of daytime airborne measurements are discussed in Ancellet and Ravetta (2003) (urban pollution in the boundary layer, ozone in the UTLS, long-range transport in the free troposphere)"

Reviewer: "Need of a rephrased conclusions showing more clearly the outcome (p 21 line 20 and 26)"

The rephrased conclusion is now:

"The purpose of this work is to provide a complete picture of Arctic ozone using measurements available during the summer 2008 POLARCAT campaigns over Canada and Greenland, where 3 aircraft were deployed and 7 ozonesonde stations intensified their ozone vertical profiling. This is the first case of such complete temporal and geographical coverage. We take advantage of the large number of airborne lidar profiles (representing 67% of the O3 measurements over Canada and 26% over Greenland). The measured ozone climatology established in this paper can also be used for future model evaluation at high latitudes. For example, in our work, while a good correspondence of the measured O3 vertical and latitudinal distribution is found with model results from WRF-Chem, a negative O3 bias of -6 to -15 ppbv between the model and the observations is found in the free troposphere over 4 km, especially over Canada. This deficiency is partly related to the WRF-Chem model stratospheric ozone initialization.

The WRF-Chem model simulation is also used to discuss the relative influence of tropospheric ozone sources at high latitude in summer. Ozone average concentrations are of the order of 65 ppbv at altitudes above 4 km both over Canada and Greenland, while they are less than 50 ppbv in the lower troposphere. For Canada, the analysis of the modeled CO distribution and the weak correlation (< 30%) of O3 and PV suggest that stratosphere-troposphere exchange (STE) is not the major contribution to tropospheric ozone at latitudes less than 70° N, where transport of North American biomass burning (BB) emissions took place during the 2008 summer. Conversely, significant STE is found over Greenland according to the better O3 versus PV correlation (> 40%) and the higher value of the 75th PV percentile. This is related to the persistence of cyclonic activity during the summer over Baffin Bay.

A weak negative latitudinal summer ozone gradient of -6 to -8 ppbv is found over Canada in the mid-troposphere between 4 and 8 km because the O3 photochemical production from BB emissions mainly takes place at latitudes less than 65° N, while STE plays a larger role at latitudes higher than 70° N. A positive ozone latitudinal gradient of 12 ppbv is observed in the same altitude range over Greenland not because of an increasing latitudinal influence of STE, but because of different long-range transport from multiple mid-latitude sources (North America, Europe and even Asia for latitudes higher than 77° N).

For the Arctic latitudes (> 80° N), free tropospheric O3 concentrations are related to a mixture of stratospheric O3 transport across the tropopause and Asian pollution, as already suggested by Roiger et al. (2011) using a case study of aircraft observations in the Arctic. Our study using more ozone measurements leads to the same conclusions."

The abstract has been also changed according to the changes made in the conclusions.

Reviewer "Need to quantify the differences between the backtrajectory analysis for zone 3 and 4 (p 20 line 16)"

The calculation of the fraction of air masses coming from North America (-150W/-60W, 50N/70N) has been added to quantify the role of the North American surface emissions on ozone in zone 3 and 4. The fractions of air masses coming from North America are 76% and 72% for zone 3 and 4, respectively. The fraction for zone 6 and 7 over Greenland is smaller (< 30%) and support the statement about the multiple mid-latitude sources controlling ozone in zone 7. This is now included in the text (p. 21 line 21-22 and p. 22 line 2-4)

Reviewer "Need to remove the sentence describing the WRF-Chem model charcateristics for aerosol modelling. (P. 7 line 18)"

Done

Please also note the supplement to this comment:
http://www.atmos-chem-phys-discuss.net/acp-2016-422/acp-2016-422-AC1-supplement.pdf

**Supplement:**

[revised manuscript text omitted]

| Zone N$^{\mathrm{o}}$ | Latitude range | Longitude range | Altitude km | $O_3$ Mean ppbv | $O_3$ Median ppbv | CO Mean ppbv | CO Median ppbv | PV 75th PVu |
|---|---|---|---|---|---|---|---|---|
| 1 | 50$^{\mathrm{o}}$/65$^{\mathrm{o}}$N | -132$^{\mathrm{o}}$/-70$^{\mathrm{o}}$W | 0-4 | 45.9 | 42.9 | 441.5 | 129.6 | - |
| 2 | 65$^{\mathrm{o}}$/80$^{\mathrm{o}}$N | -132$^{\mathrm{o}}$/-70$^{\mathrm{o}}$W | 0-4 | 38.3 | 37.7 | 71.8 | 67.8 | - |
| 3 | 50$^{\mathrm{o}}$/63$^{\mathrm{o}}$N | -132$^{\mathrm{o}}$/-70$^{\mathrm{o}}$W | 4-8 | 60.4 | 55.4 | 509.7 | 129.0 | 0.35 |
| 4 | 63$^{\mathrm{o}}$/75$^{\mathrm{o}}$N | -132$^{\mathrm{o}}$/-70$^{\mathrm{o}}$W | 4.8 | 45.7 | 47.9 | 87.8 | 87.6 | 0.31 |
| 5 | 60$^{\mathrm{o}}$/80$^{\mathrm{o}}$N | -70$^{\mathrm{o}}$/-20$^{\mathrm{o}}$W | 0-4 | 45.5 | 45.6 | 66.4 | 66.1 | - |
| 6 | 57$^{\mathrm{o}}$/65$^{\mathrm{o}}$N | -70$^{\mathrm{o}}$/-20$^{\mathrm{o}}$W | 4-8 | 51.4 | 49.0 | 83.9 | 78.8 | 0.45 |
| 7 | 65$^{\mathrm{o}}$/75$^{\mathrm{o}}$N | -70$^{\mathrm{o}}$/-20$^{\mathrm{o}}$W | 4-8 | 62.8 | 61.1 | 76.2 | 75.8 | 0.44 |
| 8 | 80$^{\mathrm{o}}$/87$^{\mathrm{o}}$N | -132$^{\mathrm{o}}$/-20$^{\mathrm{o}}$
[revised manuscript text omitted]

---

## Author Comment (AC2) · 23 Sep 2016

We thank reviewer 2 for his review. We apologize for the large number of English errors and the new manuscript has been carefully edited by one of the English speaking co-author. Find below the answers to the questions raised by the reviewer. See also attachment in the Reply to Reviewer 1 for the new revised manuscript.

Reviewer :"The paper provides a detailed description of tropospheric ozone measurements during the POLARCAT campaign in summer 2008, and a series of analysis of its variability and possible sources affecting the variability by using WRF-Chem chemistrytransport model. The description is intensive and the analysis is generally sound, reaching some interesting conclusions. I found the paper be a nice piece of work contributing to better understanding of sources and processes affecting the summertime ozone at high-latitudes. The paper is well within the scope of ACP, and can eventually be published. However, I found some descriptions/illustrations are redundant. There are many errors in English. Some figures look preliminary. These made me difficult to follow what the authors are trying to tell us. So, I would suggest some technical suggestions that should be addressed before publication. Figures 5, 7, and 9: The same data are plotted in both linear and logarithmic scales. I doubt if the authors really need logarithmic plots, as they do not discuss much on the log plots. The log plots seem redundant to me."

The log plots have been removed as suggested by the reviewer because they are not specifically discussed in the text. We also agree with the reviewer that some plots needed improvement. The measured/modeled ozone scatterplot (Figure 5) has been changed to better distinguish tropospheric and stratospheric data (now in white) and to include the O3 concentration unit in the x-axis label. The back trajectory plots (Fig. 11 and 12) include now the name of the corresponding zone shown in Fig. 6 and 8 and unit for the altitude color scale. A new figure is created (Fig. 10) for the map of the MODIS aerosol optical depth and the MAP of the CFS fire counts distribution has been added to it

Reviewer : "There are many errors in English: L1: The goals of the paper are ... L7: Ozone, CO and . . . is too much detail in Abstract, and can be removed. L10: The average ozone concentrations are 65 ppbv ... L14: ... modeled CO ... L18: ... ozone gradient of -6 to -8 ppbv . . . P7, L17: MEGAN (Model of ...) Figure 2 caption: Intercomparison of ozone measurements ..."

We thank the reviewer for his help in the manuscript editing. The English-speaking co-authors have also corrected many grammar errors.

Reviewer: "Captions in other figures: Measured O3 (ppbv), Modeled O3 (ppbv) - need units!"

Done in Fig. 5, 7, 9

Reviewer: "Figures 3 and 4: I would make difference plots between observation and model, to illustrate where in height and latitude the model is good or bad. This is not necessary but please consider."

We understand the reviewer remark but the measurement/model comparison does not focus on small scale differences. The goal of the comparison is to check that the main latitudinal and vertical gradients are well reproduced in the WRF-Chem simulations. This is why we do not wish to produce a detailed 2-D plot of the modeled/mesuared ozone differences which will show mainly model/measurement spatial/temporal mismatches. We believe that the differences can be better discussed using two different 2D ozone plots for the model and the data and the detailed statistical informations about model /measurement differences being provided in table 5 for each region selected for the vertical and latitudinal gradient analysis.

---

## Author Response (AR2)

Co-Editor Decision: Publish subject to technical corrections (11 Oct 2016) by Eliza Harris

Comments to the Author:

- Abstract L3: spell out numbers under 10 ie. three aircraft and seven ozonesonde... (throughout manuscript, except dates and table/fig crossreferences)

done

- Figure 1: Are these different flight tracks? Can you make each different flight a different colour or different shade of blue?

Done (color for each aircraft)

- Figure 2: Plot the altitudes as the correct values (not scaled by 50/100) on the right hand y axis please, instead of scaled and bunched with O3 on the left hand axis!

Done

- Figures 3 and 4: Reviewer 2 suggested a difference plot between measurements and model. The authors believe that the information in Table 5 is sufficient. I would tend to agree with the reviewer; a full 2D latitudinal difference plot may not be necessary, as it will emphasise spatial differences only as mentioned by the authors, but some visual representation of match/mismatch would be very useful. Can you plot altitude profiles at different latitudes to compare or something else, that will visually show the main points of agreement/disagreement?

We finally decided to include the altitude/latitude relative difference plot (now called Fig. 5) as suggested by the reviewer 2 and the editor. The discussion of the modeled versus measured ozone comparision in section 4 is slightly updated to include the discussion of this figure (p. 9 line 11-12)

- Page 6 L4: space between Table 3 - check this throughout, there are several locations with no space between Fig. and Table and the following number.

Done

- Figure 7 and 9 (now 8 and 10), y axis label: "km" not "Km"

Done

- Figure 10 (now 11): It is hard to compare. Can you rotate the MODIS view so the map of Canada has the same orientation as the forest fire counts? Similarly for Figure 11 and 12 (now 12 and 13).

We did not rotate the maps because we cannot do it with our plotting routine. Since it is not a strong requirement figures 11 to 13 are unchanged.